# Identifying Mortality Predictors in Hospitalized COVID-19 Patients: Insights from a Single-Center Retrospective Study at a University Hospital

**DOI:** 10.3390/microorganisms12051032

**Published:** 2024-05-20

**Authors:** Ondrej Zahornacky, Alena Rovnakova, Maria Surimova, Stefan Porubcin, Pavol Jarcuska

**Affiliations:** 1Department of Infectology and Travel Medicine, Faculty of Medicine, Louis Pasteur University Hospital, Pavol Jozef Šafarik University, 041 90 Košice, Slovakia; ondrejzahornacky@gmail.com (O.Z.); alenarovnakova@gmail.com (A.R.); 2Institute of Mathematic, Faculty of Science, Pavol Jozef Šafarik University, 041 90 Košice, Slovakia; maria.surim@gmail.com

**Keywords:** COVID-19, risk factors, mortality, predictors, hospitalized

## Abstract

Introduction: The pandemic instigated by the SARS-CoV-2 virus has led to over 7 million deaths globally, primarily attributable to viral pneumonia. Identifying fundamental markers associated with an elevated risk of mortality can aid in the early identification of patients prone to disease progression to a severe state, enabling prompt intervention. Methods: This was a single-center, retrospective study. Results: In this study, we examined 299 patients admitted to the Department of Infectology and Travel Medicine in Košice, Slovakia, with PCR-confirmed COVID-19 pneumonia. Patients were monitored from 1 January 2021 to 31 March 2021, with the endpoint being discharge from the hospital or death. All patient-related data were retrospectively collected from medical records. This study identified several risk factors significantly associated with an increased risk of mortality, including the requirement of HFNO (*p* < 0.001), age over 60 years (*p* < 0.001), Ne/Ly values of >6 (*p* < 0.001), as well as certain lymphocyte subtypes—CD4+ < 0.2 × 10^9^/L (*p* = 0.035), CD8+ < 0.2 × 10^9^/L (*p* < 0.001), and CD19+ < 0.1 × 10^9^/L (*p* < 0.001)—alongside selected biochemical inflammatory markers—IL-6 > 50 ng/L (*p* < 0.001) and lactate > 3 mmol/L (*p* < 0.001). Conclusions: We confirmed that the mentioned risk factors were significantly associated with the death of patients from viral pneumonia in the hospital.

## 1. Introduction

COVID-19, caused by the SARS-CoV-2 virus, is a highly transmissible viral illness characterized by a diverse clinical course. Infections can manifest as asymptomatic cases or mild respiratory symptoms or can progress to severe viral pneumonia with a risk of respiratory failure, often driven by cytokine storms, a fundamental pathophysiological mechanism. The initial cases of this disease emerged in December 2019 in Wuhan, Hubei Province, China. Subsequently, the infection rapidly spread worldwide. Due to the escalating global transmission, the World Health Organization (WHO) declared the situation a global pandemic on 11 March 2020 [1,2].

Various hematological, biochemical, and immunological parameters play a crucial role throughout the course of COVID-19, particularly concerning disease prognosis and the onset of complications [3]. 

The systemic inflammatory response, characterized primarily by elevated levels of cytokines such as IL-1 and IL-6, plays a pivotal role in the pathogenesis of COVID-19, potentially culminating in severe complications and mortality [4,5].

During acute hypoxia, there is an augmented conversion of pyruvate into lactic acid (lactate) within tissues. The cytokine storm associated with acute respiratory distress syndrome (ARDS) exacerbates hypoxemia in the body, further worsening lactic acidosis [6].

Elevated serum lactate levels may serve as a predictor of increased mortality risk, a need for vasopressors, and a requirement for mechanical ventilation (MV), leading to intensive care unit admission in patients with advanced viral pneumonia [7,8].

Changes in the total lymphocyte count, particularly lymphopenia, and alterations in individual lymphocyte subtypes are fundamental laboratory parameters associated with a severe prognosis in hospitalized patients. Among these, subtypes of CD4+ and CD8+ T lymphocytes are regarded as significant prognostic biomarkers of the disease [9,10,11].

For these reasons, our study focused on conducting a retrospective analysis of selected hematological, biochemical, and immunological parameters to prognosticate the outcomes of patients with viral pneumonia.

## 2. Materials and Methods

### 2.1. Study Design and Participants

A retrospective observational study was conducted from 1 January 2021 to 31 March 2021. During this period, a total of 299 patients with RT-PCR-confirmed COVID-19 infection (nasopharyngeal swab) were hospitalized at the Department of Infectology and Travel Medicine in Košice, Slovakia.

### 2.2. Inclusion Criteria

The cohort comprised all patients, both men and women, aged 19 years and above, who tested positive for COVID-19 via RT-PCR and had ultrasound-confirmed viral pneumonia. Other potential etiologies of the disease, such as influenza and adenovirus, were excluded through PCR examination. All patients necessitating hospitalization for viral pneumonia, irrespective of whether they required oxygen supplementation, were included in this study.

### 2.3. Exclusion Criteria

Patients under the age of 19, individuals without confirmed viral pneumonia, and COVID-19-positive patients requiring hospitalization for reasons other than pneumonia were excluded from this study.

The final endpoint of follow-up was either discharge from the hospital or death. For the analysis, patients were divided into two groups: survivors and non-survivors. The following statistical data were processed: age; sex; body mass index (BMI); the necessity of oxygen treatment including low-flow oxygen therapy (LFO), high-flow oxygen therapy (HFNO), or MV; length of hospitalization (in days); mortality; the nadir values of absolute CD4+, CD8+, and CD19+ T lymphocytes and NK cells; the nadir levels of lactate and interleukin 6 (IL-6); and the nadir ratio of absolute neutrophils to lymphocytes (Ne/Ly). The results of the monitored laboratory parameters were obtained from the hospital’s information system.

### 2.4. Statistical Analysis

The baseline characteristics of the cohort are presented as means, frequencies, and percentages. The relative risk (RR), also known as the risk ratio, was calculated by comparing the incidence of a risk factor (e.g., age over 60 years, gender, obesity, etc.) in the exposed group (e.g., patients over 60 years) to the incidence of the risk factor in the non-exposed group (e.g., patients under 60 years).

For the statistical analysis, we utilized the openEpi software (version 3.01) and the TwoByTwo table to calculate the odds ratio (OR). Fisher’s exact test and the Mann–Whitney U test were employed to determine the *p*-value of statistical significance. A *p*-value of less than 0.05 was considered statistically significant.

## 3. Results

During the period from 1 January 2021, to 31 March 2021, a total of 299 patients with RT-PCR-confirmed COVID-19 disease and ultrasound-verified viral pneumonia were admitted to the Department of Infectious Diseases and Travel Medicine in Košice, Slovakia (Table 1). The cohort consisted of 157 (52.51%) men and 142 (47.49%) women, with an average age of 61 years (range, 26–92 years), an average weight of 87.82 kg (range, 44–151 kg), and an average BMI of 30.06 (range, 13.8–49.31). The average length of hospitalization was 10.5 days (range, 2–53 days). Among the patients, 270 (90.30%) required oxygen treatment in various forms, with 31 (24.08%) experiencing progression to respiratory insufficiency requiring HFNO, while 41 patients required HFNO immediately upon hospital admission. MV was necessary in 17 patients (5.69%) due to the progression of respiratory insufficiency, and 9 patients (3.01%) required the continuation of oxygen therapy at home via long-term oxygen therapy (LTO). Out of the hospitalized patients, 62 (20.74%) died.

In the monitored group, the average absolute value of CD4+ T lymphocytes (Table 2) was 0.47 × 10^9^/L (range: 0.02–1.86), CD8+ T lymphocytes 0.26 × 10^9^/L (range: 0.02–1.68), CD19+ T lymphocytes 0.18 × 10^9^/L (range: 0–0.94), and NK (natural killer) cells 0.20 × 10^9^/L (range: 0–0.73). The average value of IL-6 was 158.3 ng/L (range: 0.02–2700), lactate 2.41 mmol/L (range: 0.67–11.3), and the Ne/Ly ratio 13.33 (range: 0.19–143). 

Statistically significant differences were observed between the two monitored groups (survivors/non-survivors). Table 3 summarizes the differences between the two observed groups. The analyzed results indicate that surviving patients had a statistically significantly longer duration of hospitalization, by 2.5 days on average compared with deceased patients (*p* < 0.001). Surviving patients also had a longer duration of oxygen treatment with LFO by an average of 0.8 days (*p* = 0.005) and HFNO by an average of 1.5 days (*p* = 0.001). However, no significant difference was found in the duration of MV (*p* = 0.483).

Regarding immunological parameters, statistically significant differences were observed between the monitored groups in the number of CD4+ T lymphocytes (*p* < 0.001), CD8+ T lymphocytes (*p* < 0.001), CD19+ T lymphocytes (*p* = 0.001), and the Ne/Ly index (*p* = 0.001). No significant difference was observed in the case of NK cells (*p* = 0.789). Additionally, a statistically significant difference was observed in the levels of IL-6 (*p* < 0.001) and lactate (*p* < 0.001). The box plot in [Fig microorganisms-12-01032-ch001] effectively summarizes the distribution of lymphocyte counts in non-surviving and surviving COVID-19 patients, highlighting the potential association between lower lymphocyte levels and increased mortality.

Within the monitored groups of patients, various risk factors and their influences on mortality were analyzed. Table 4 summarizes the monitored risk factors, their demographics, and their relationship with the mortality of hospitalized patients. Also the monitored risk factors, and their relationship with the mortality of hospitalized patients is described in [Fig microorganisms-12-01032-ch002] and [Fig microorganisms-12-01032-ch003]. In the case of oxygen treatment, we found that LFO was not a risk factor for death (*p* = 0.339; RR = 0.73; OR = 0.66), nor was MV (*p* = 0.999; RR = 0.84; OR = 0.81). However, oxygen therapy with HFNO proved to be a significant risk factor for death (*p* < 0.001; RR = 5.73; OR = 11.65).

Other risk factors associated with an increased risk of death included an age over 60 years (*p* = 0.001; RR = 47.57; OR = 74.11), a CD4+ T lymphocyte count below 0.2 × 10^9^/L (*p* = 0.035; RR = 1.71; OR = 2.05), a CD8+ T lymphocyte count below 0.2 × 10^9^/L (*p* < 0.001; RR = 2.99; OR = 3.87), a CD19+ T lymphocyte count of less than 0.1 × 10^9^/L (*p* < 0.001; RR = 3.27; OR = 4.54), and an NK cell count of less than 0.05 × 10^9^/L (*p* = 0.024; RR = 2.18; OR = 3.04).

Among the selected inflammatory parameters, the following were identified as risk factors associated with the death of hospitalized patients: IL-6 levels above 50 ng/L (*p* < 0.001; RR = 7.96; OR = 10.89), lactate levels above 3 mmol/L (*p* < 0.001; RR = 3.13; OR = 4.76), and Ne/Ly indexes above 6 (*p* = 0.001; RR = 8.47; OR = 11.34).

## 4. Discussion

The primary objective of this retrospective observational study was to identify the underlying risk factors associated with mortality from SARS-CoV-2 pneumonia during the pandemic caused by the Alpha variant of the virus. Within the monitored group, we analyzed a total of 299 patients hospitalized at the Department of Infectology and Travel Medicine in Košice, Slovakia, with PCR-confirmed COVID-19 infection and ultrasound-verified viral pneumonia during the study period. All patients underwent exclusion of other etiologies of interstitial pneumonia through PCR testing. It is worth noting that the study period was characterized by a high incidence of the Alpha variant of the SARS-CoV-2 virus in Slovakia [12]. In the results of Guarino et al., up to 21% of patients hospitalized with viral pneumonia during the first wave of the pandemic in 2020 also had a bacterial superinfection (caused by *Mycoplasma pneumoniae* and *Chlamydia pneumoniae*). However, these authors did not observe a higher mortality rate in co-infected patients [13].

In the monitored group, 62 patients succumbed to the disease, constituting 20.74% of the total number of hospitalized patients. The in-hospital mortality in patients with SARS-CoV-2 infection was variably high across several international studies. The estimate of in-hospital mortality during the first months of the outbreak in 2020 was 17% in a meta-analysis of 33 studies between January and April 2020, and the mortality was lower in general inpatients (11%) compared with critically ill patients (40%) [14]. However, these reports unraveled a degree of heterogeneity with geographical differences, and a mortality of up to 31% was reported, as in the cases of 522,167 patients hospitalized with SARS-CoV-2 infection in Brazil by December 2020 [15]. Such a diversity of results might reflect the virulence of SARS-CoV-2 in different populations, as well as the local capacity of the national healthcare systems. Comparatively, according to the findings of other authors, the mortality rate among hospitalized patients with the Alpha variant of the SARS-CoV-2 disease typically ranges from 6% to 9% [16,17,18]. 

The disparity in mortality rates between our patients and the global data may primarily stem from the composition of our study group, which predominantly consisted of obese patients, with an average BMI of 30.06 (ranging from 13.8 to 49.31). Additionally, the increased mortality could also be attributed to the unavailability of intravenous remdesivir during the study period. Furthermore, oral antivirals such as molnupiravir and nirmatrelvir/ritonavir were not accessible at that time, which might have impacted treatment outcomes.

The data presented in Table 3 illustrate statistically significant differences between the two monitored groups of patients (survivors/deceased), particularly in the length of hospitalization. Surviving patients were hospitalized longer, by 1.5 days on average (*p* < 0.001). Similarly, there were notable differences in the duration of oxygen treatment in the form of LFO, with surviving patients receiving LFO for a longer period (*p* = 0.005). However, it was not established that LFO is a risk factor associated with a higher risk of death (*p* = 0.339).

Conversely, in the case of the duration of HFNO, deceased patients had a longer duration of HFNO compared with survivors (*p* < 0.001). Moreover, HFNO was identified as a significant risk factor associated with more than five times the risk of death (*p* < 0.001; RR = 5.73; OR = 11.65). This observation is likely related to the fact that patients who required HFNO and subsequently failed to respond were subsequently transferred to MV. Patients were typically maintained on HFNO only if further escalation of oxygen treatment was deemed inappropriate, primarily due to their overall poor prognostic status.

No statistical difference was demonstrated in the length of MV (*p* = 0.483) and in gender between the observed groups (*p* = 0.557). The role of the male gender is controversial. Sex differences in COVID-19 case fatality were observed in the first half of 2020, and male gender seemed to be associated with poor outcomes [19]. However, this association was inconsistently reported across studies [20]. This apparent gender gap was hypothesized to reflect other factors such as the access to health services and its representation in hospital settings, rather than biological differences in response to SARS-CoV-2 pathology.

Similarly, the authors Saito et al. did not note differences in mortality depending on gender. It is noteworthy that the authors did not find differences in the length of oxygen treatment between the monitored groups [21]. 

According to the authors Brainard et al., male gender is identified as a risk factor associated with a higher risk of death [22]. 

Age consistently emerges as a prominent predictor of mortality across various COVID-19 variants, including the Alpha variant. Research indicates that hospitalized patients above a certain age threshold, often 65 or older, have a significantly elevated risk of succumbing to the infection compared with younger individuals. This increased vulnerability is believed to stem from age-related decline in immune function and the higher prevalence of underlying medical conditions [23,24].

Likewise, the conclusions of the authors Corradini et al. and Moreno-Torres et al. confirm that older age (>60 years) and the presence of comorbidities are independent predictors of severe outcomes in hospitalized patients with SARS-CoV-2 infection, in line with other case studies from Spain and Italy [25,26]. Similarly, in our study, age over 60 was identified as a risk factor associated with a higher risk of death (*p* < 0.001; RR = 47.57; OR = 74.11).

Multiple studies have established a clear link between BMI and an increased risk of mortality in patients hospitalized with the Alpha variant of COVID-19. Individuals with a BMI categorized as obese (over 30) were observed to have a significantly higher risk of hospitalization, critical illness, and death from the Alpha variant compared with those with a healthy weight. The development of complications in obese patients could be explained by various pathophysiological mechanisms, including increased expression of angiotensin-2 in adipose tissue, chronic inflammation, and the amplification of the pro-inflammatory response, in addition to endothelial damage and hypercoagulability [27].

However, in the case of our study, a BMI over 30 was not confirmed as a risk factor for death (*p* = 0.229; RR = 1.35; OR = 1.47). This discrepancy may be attributed to the fact that the patients enrolled in this study had an average BMI of 30.06 (ranging from 13.8 to 49.31). Notably, 127 patients (42.47%) of the total number had a BMI above 30, indicating that a large proportion were obese patients [28,29,30].

We also observed significant differences in monitored immunological parameters, specifically in the nadir values of CD4+, CD8+, and CD19+ T lymphocytes between the mentioned groups of patients. Surviving patients exhibited a statistically significantly higher number of CD4+ (*p* < 0.001), CD8+ (*p* < 0.001), and CD19+ T lymphocytes (*p* < 0.001) compared with the group that died. However, in the case of NK cells, no statistically significant difference was found (*p* = 0.789).

Similar findings have been reported by other authors, who have demonstrated that low numbers of CD4+, CD8+, and CD19+ T lymphocytes are associated with a higher risk of death. The number of CD4+ T lymphocytes is currently considered an important prognostic marker of survival [31,32,33]. 

Similar conclusions were also confirmed in our study. A low number of CD4+ T lymphocytes (nadir < 0.2 × 10^9^/L) was associated with an almost doubled risk of death (*p* = 0.035; RR = 1.71; OR = 2.05). Furthermore, in the case of CD8+ T lymphocytes (nadir of less than 0.1 × 10^9^/L), we observed an even stronger correlation (*p* < 0.001; RR = 2.99; OR = 3.87) with the risk of death. Notably, the number of CD19+ T lymphocytes (nadir of less than 0.1 × 10^9^/L) exhibited the strongest correlation (*p* < 0.001; RR = 3.27; OR = 4.54) with patient death. These findings underscore the importance of these lymphocyte subtypes as prognostic markers in COVID-19 patient outcomes.

An exaggerated inflammatory response of the body plays a central role in the pathogenesis of COVID-19. Numerous publications have confirmed that the cytokine storm, characterized by elevated levels of cytokines in the blood, is implicated in the progression of the disease to a severe state and is a significant prognostic factor associated with death. This is exemplified by the conclusions drawn by Li et al. and Shi et al. [34,35]. 

Levels of these markers reflect the intensity of the cytokine-mediated hyperinflammatory response and appear to be strongly associated with increased mortality. In our study, we analyzed the nadir values of IL-6 and its effect on the mortality of patients in the cohort. We found that patients with a plasma value of IL-6 of > 60 ng/L had a statistically significant, almost eight times higher risk of death (*p* < 0.001; RR = 7.96; OR = 10.89). Additionally, a meta-analysis based on a total of nine studies, including 1426 patients, suggested a cut-off value of 55 ng/mL to identify severe forms [36].

Regarding mortality, only a few studies have analyzed the predictive value of IL-6 to assess critical and fatal forms. The study conducted by Herold et al. defined a value of 80 ng/mL as indicative of the need for mechanical ventilation, with a median time of 1.5 days [37]. 

Ne/Ly combines two parameters: the ratio of the number of absolute neutrophils and the number of absolute lymphocytes. An increase in the number of neutrophils reflects the intensity of systemic inflammation, while lymphopenia reflects the sequestration of lymphocytes at the site of inflammation and their apoptosis. In our ratio study, we demonstrated that a nadir value of Ne/Ly above 6 is associated with a statistically significant, almost ninefold higher risk of death (*p* < 0.001; RR = 8.47; OR = 11.34). Similar findings were published in a meta-analysis by Simadibrat et al., which indicated that patients with severe COVID-19 disease and non-survivors had higher Ne/Ly values on admission than survivors. These results were consistent with the findings of Kudlinsky et al. and Sayah et al. [38,39,40].

Lactate is a metabolite that indicates the anaerobic state of a cell, often attributed to tissue destruction or certain medications such as metformin and acetaminophen. Our analysis revealed statistically significant differences in serum lactate levels, with a mean of 12.15 mmol/L in the deceased group compared with a mean of 3.27 mmol/L in the surviving group. We found that a lactate value of 3 mmol/L is associated with a statistically significant higher risk of death (*p* < 0.001; RR = 3.13; OR = 4.76). Similarly, Pouyan et al. reported a significant relationship between lactate levels and the mortality of hospitalized patients [41].

In a study conducted by Isler et al., and similarly reported by Carpenè et al., lactate levels can even serve as a therapeutic target to evaluate the effects of drugs on patients with COVID-19 [42,43].

This study had several limitations. First, the monocentric nature of the data source and analysis may reflect regional and local health policies but might not be generalized to all hospital settings at a country level. Second, data on associated comorbidities and the extent of lung involvement based on computed tomography findings were not analyzed, mainly because computer tomography of the lungs was not performed in every patient included in this study.

## 5. Conclusions

The main goal of this study was to identify risk factors and determine their causal relationship with mortality in patients hospitalized with COVID-19 viral pneumonia at the Department of Infectious Diseases and Travel Medicine in Košice, Slovakia during the Alpha wave of the pandemic. As part of this analysis, we confirmed that the duration of oxygen treatment in the form of HFNO (*p* < 0.001), age over 60 years (*p* < 0.001), Ne/Ly values of >6 (*p* < 0.001), selected lymphocyte subtypes—CD4+ < 0.2 × 10^9^/L (*p* = 0.035), CD8+ < 0.2 × 10^9^/L (*p* < 0.001), and CD19+ < 0.1 × 10^9^/L (*p* < 0.001)—as well as selected biochemical inflammatory markers—IL-6 > 50 ng/L (*p* < 0.001) and lactate > 3 mmol/L (*p* < 0.001)—are significant risk factors associated with the death of patients from viral pneumonia in hospitals.

## Data Availability

The raw data supporting the conclusions of this article will be made available by the authors on request.

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
