# Peer review of "Identifying Mortality Predictors in Hospitalized COVID-19 Patients: Insights from a Single-Center Retrospective Study at a University Hospital"

_microorganisms, 2024, doi:10.3390/microorganisms12051032_

Round 1

Reviewer 1 Report

Comments and Suggestions for Authors

The paper sounds 

1-But i will review from my statistical point of view 

(p < 0.001), age over 60 years (p < 0.001) 

what if the p-value is 0.005, why do you used 0.001??

2- Some graphs should be added to elaborate the results concluded 

3- Regarding Table 1, you depended on some factors; I think those factors are too much. Some of them could be removed 

4-  How do you calculate the P- P-value in Table 3?

5- Add comments on table 4 and 3 to elaborate the findings

6- A new related references should be added 

7- Some mistakes in the paper should be corrected  see the conclusion as an example 

Comments on the Quality of English Language

Good language 

Author Response

Dear reviewer, thank you for your recommendations. We added all your comments to our new version of manuscript. We send the answers to your questions below.

1-But i will review from my statistical point of view  (p < 0.001), age over 60 years (p < 0.001) what if the p-value is 0.005, why do you used 0.001??

This result means that the p value acquired a value smaller than 0.001, mostly it was much smaller numbers, e.g. 0.0000000001. That is why we decided to round these small values to 3 decimal places.

2- Some graphs should be added to elaborate the results concluded

   We added three graphs to our manuscript.

3- Regarding Table 1, you depended on some factors; I think those factors are too much. Some of  them could be removed

We corrected Table 1 as recommended.

4-  How do you calculate the P- P-value in Table 3?

The P-values were calculated using t-test, Fisher’s exact test for normally distributed data and Mann-Whitney test was used after Wilk-Shapiro test of normality failed, with 2 × 2 contingency tables – added to text.

5- Add comments on table 4 and 3 to elaborate the findings

added comments

6- A new related references should be added

We added a 7 new related references to manuscript.

7- Some mistakes in the paper should be corrected  see the conclusion as an example

We corrected mistakes.

Best regards, authors 

Reviewer 2 Report

Comments and Suggestions for Authors

Proposed paper is interesting and well written, however, some revisions are needed before it can be accepted for publication:

- I found a similar study with similar findings (10.1007/s10389-021-01675-y). Authors can consider to cite it and discuss differences and similarities in the discussion section.

- Are there any data on atrial fibrillation and other arryhtmias? this is an important issue of COVID-19 infection that has been also related to a worst outcome (10.3390/biomedicines10081940). 

HFNO should be explained in the ABS.

- Are other data such as comorbidities and computer tomography evaluated? if not this should be stated as a strong limitation of the study.

Author Response

Dear reviewer, thank you for your recommendations. We added all your comments to our new version of manuscript. We send the answers to your questions below.

- I found a similar study with similar findings (10.1007/s10389-021-01675-y). Authors can consider to cite it and discuss differences and similarities in the discussion section.

We compared our results with the results of the mentioned study

- Are there any data on atrial fibrillation and other arryhtmias? this is an important issue of COVID-19 infection that has been also related to a worst outcome (10.3390/biomedicines10081940).

In our study, we did not monitor the presence of atrial fibrillation.

HFNO should be explained in the ABS.

  • explained

- Are other data such as comorbidities and computer tomography evaluated? if not this should be stated as a strong limitation of the study.

Data on associated comorbidities and the extent of lung involvement based on computed tomography findings were not analyzed, mainly because computer tomography of lungs was not performed in every patient included in the study. We have added this information to the text as limitations of the paper.

Best regards, atuhors

Round 2

Reviewer 2 Report

Comments and Suggestions for Authors

Authors replies to all the query raised and paper can now be accepted for publication.

Author Response

Thank you